# Hepatic lipase (*LIPC*) sequencing in individuals with extremely high and low high-density lipoprotein cholesterol levels

Dilek Pirim[1,2], Clareann H. Bunker[3], John E. Hokanson[4], Richard F. Hamman[4], F. Yesim Demirci[1], M. Ilyas Kamboh[1]*

1 Department of Human Genetics, Graduate School of Public Health, University of Pittsburgh, Pittsburgh, Pennsylvania, United States of America, 2 Department of Molecular Biology and Genetics, Faculty of Arts & Science, Bursa Uludag University, Gorukle, Bursa, Turkey, 3 Department of Epidemiology, Graduate School of Public Health, University of Pittsburgh, Pittsburgh, Pennsylvania, United States of America, 4 Department of Epidemiology, Colorado School of Public Health, University of Colorado Denver, Aurora, Colorado, United States of America

* kamboh@pitt.edu

**Data Availability Statement:** All relevant data are within the paper and its Supporting information files.

## Abstract

Common variants in the hepatic lipase (*LIPC*) gene have been shown to be associated with plasma lipid levels; however, the distribution and functional features of rare and regulatory *LIPC* variants contributing to the extreme lipid phenotypes are not well known. This study was aimed to catalogue *LIPC* variants by resequencing the entire *LIPC* gene in 95 non-Hispanic Whites (NHWs) and 95 African blacks (ABs) with extreme HDL-C levels followed by *in silico* functional analyses. A total of 412 variants, including 43 novel variants were identified; 56 were unique to NHWs and 234 were unique to ABs. Seventy-eight variants in NHWs and 89 variants in ABs were present either in high HDL-C group or low HDL-C group. Two non-synonymous variants (p.S289F, p.T405M), found in NHWs with high HDL-C group were predicted to have damaging effect on LIPC protein by SIFT, MT2 and PP2. We also found several non-coding variants that possibly reside in the circRNA and lncRNA binding sites and may have regulatory potential, as identified in rSNPbase and RegulomeDB databases. Our results shed light on the regulatory nature of rare and non-coding *LIPC* variants as well as suggest their important contributions in affecting the extreme HDL-C phenotypes.

## Introduction

The hepatic lipase (*HL*) gene, also known as *LIPC*, is one of the member of the lipase gene family that functions in the uptake of high density lipoprotein cholesterol (HDL-C) and plays an essential multifunctional role in lipoprotein and lipid metabolism [1–3]. The *LIPC* gene is located on chromosome 15q21-q23 spanning approximately 138kb and consists of a total 9 exons coding 499 amino acid (~55kDa) glycoprotein HL or LIPC [4–6]. Due to its critical role in lipid metabolism, the LIPC activity dramatically causes fluctuating in plasma lipoprotein-lipid levels [7–9]. Previous studies have shown genetic variants in the *LIPC* affect the activity of LIPC and contribute to the risk of several diseases, including coronary artery disease (CAD), type 2 diabetes, metabolic syndrome and HL deficiency. Genome-wide association

**Funding:** This study was supported by the National Heart, Lung and Blood Institute (NHLBI) grant, HL084613 (M. Ilyas Kamboh). The funders had no role in study design, data collection and analysis, decision to publish, or preparation of the manuscript. There was no additional external funding received for this study.

**Competing interests:** The authors have declared that no competing interests exist.

studies (GWASs) have revealed several common [minor allele frequency (MAF>5%)] *LIPC* variants that were associated with plasma levels of HDL-C, triglycerides (TG), and total cholesterol (TC) [10–19]. In addition to common variants, rare and low frequency variants (MAF≤5%) also impact phenotypic variation in plasma lipid levels [20]. Thus, it is imperative to identify and elucidate the roles of rare variants in lipid genes to explain the missing heritability of lipid phenotypes. Indeed, resequencing individuals with extreme lipid profiles has shown to confer advantage for detecting rare variants contributing to plasma lipid variation [21–27], which warrant further investigation in diverse population.

In this study, we resequenced the *LIPC* gene in 190 subjects with extreme HDL-C levels in order to identify the distribution of *LIPC* sequence variants in extreme phenotypes as well as to assess their functional relevance.

## Materials and methods

### Subjects

The study sample comprised of 190 samples, including 95 African Blacks (ABs) and 95 non-Hispanic Whites (NHWs) with extreme HDL-C levels. These small subset of samples were selected from the upper and lower 10th percentile of HDL-C distribution from two epidemiological well-characterized samples including 623 NHWs and 788 ABs for variant discovery purposes [28–31]. The biometric and quantitative data of 95 NHW and 95 AB samples with extreme HDL-C groups are presented in Table 1. The research was conducted in accordance with the relevant ethical guidelines/regulations and approved by the Institutional Ethical Review Boards of the University of Pittsburgh and University of Colorado Denver. Written informed consent was obtained from all participants.

**Table 1. Biometric and quantitative data (mean± SD) of the 95 NHW and 95 African black samples with extreme\* HDL-C levels.**

| | NHWs (n = 95) | | | Entire Sample (n = 623) | African blacks (n = 95) | | | Entire Sample (n = 788) |
|---|---|---|---|---|---|---|---|---|
| | High HDL-C (n = 47) (HDL-C range: 58–106 mg/dL) | Low HDL–C (n = 48) (HDL-C range: 20–40 mg/dL) | *P-*value | | High HDL-C (n = 48) (HDL-C range: 68.30–99 mg/dL) | Low HDL-C (n = 47) (HDL-C range: 10.30–35 mg/dL) | *P-*value | |
| Sex (M/F) | 24/23 | 24/24 | 1 | 295/328 | 24/24 | 23/24 | 1 | 493/293 |
| Age (years) | 55.45 ± 9.80 | 53.03 ± 10.54 | 0.25 | 52.83 ± 11.41 | 41.29 ± 8.72 | 40.87 ± 7.12 | 0.80 | 40.95 ± 8.39 |
| BMI (kg/m²) | 23.17 ± 3.17 | 27.35 ± 3.90 | 1.2E-07 | 25.51 ± 4.06 | 22.06 ± 4.70 | 23.91 ± 5.51 | 0.08 | 22.87 ± 4.04 |
| TC (mg/dl) | 227.34 ± 51.76 | 208.81± 44.65 | 0.07 | 216.99 ± 43.55 | 201 ± 39.68 | 141.68 ± 31.03 | 2.4E-12 | 172.01 ± 38.47 |
| LDL-C (mg/dl) | 126.84 ± 46.95 | 125.54 ± 54.97 | 0.90 | 136.99 ± 40.80 | 112.55 ± 39.75 | 95.04 ± 28.28 | 0.02 | 109.25 ± 34.40 |
| HDL-C (mg/dl) | 77.68 ± 13.32 | 31.81 ± 4.37 | 2.2E-16 | 50.76 ± 14.35 | 76.05 ± 7.53 | 25.51 ± 5.66 | 2.2E-16 | 47.88 ± 12.87 |
| TG (mg/dl) | 114.09 ± 60.88 | 240.21 ± 153.22 | 1.7E-06 | 142.72 ± 93.49 | 61.98 ± 19.85 | 95.79 ± 73.21 | 0.004 | 72.96 ± 39.32 |
| ApoB(mg/dl) | 87.88 ± 25.49 | 89.61± 25.18 | 0.80 | 149.62 ± 33.33 | 66.00 ± 20.22 | 69.64 ± 21.46 | 0.40 | 66.98 ± 22.19 |
| ApoA1 (mg/dl) | 174.08 ± 3.57 | 130.20 ± 2.78 | 1.4E-06 | 87.72 ± 24.27 | 166.04 ± 28.19 | 103.84 ± 27.23 | 2.2E-16 | 137.03 ± 28.46 |

P-values were calculated based on the original values by using t-test. No covariates were included.

\*Adjusted for sex and age: High/Low HDL-C groups correspond to ≥90th % tile and ≤10th % tile of the HDL-C distribution.

TC: Total cholesterol; LDL-C: Low-density lipoprotein cholesterol; HDL-C; high-density lipoprotein cholesterol; TG: Triglycerides.

## Lipid measurements

Friedewald equation was used to calculate the plasma low-density lipoprotein cholesterol (LDL-C) levels and esterase-oxidase method was employed for measuring fasting total cholesterol levels [32, 33]. Enzymatic procedures described in Harris et al. [34] were used to determine serum HDL-C and triglyceride concentrations.

## DNA sequencing

The entire *LIPC* gene (NG_011465, NM_000236) (excluding the large intron 1) (see S1 Fig), plus 1kb in 3' and 5' flanking sequence comprising ~ 33kb was sequenced in both direction using a total of 46 resequencing amplicons. DNA isolations were conducted using buffy coat and blood clots from of NHW and AB samples, respectively, following the standard DNA isolation protocols. PCR primers were prepared by using the Primer 3 software (http://frodo.wi.mit.edu/primer3/) (see S1 Table). PCR conditions are available upon request. Sanger sequencing was performed on the ABI 3730x1DNA analyzers in a commercial lab (Beckman Coulter Genomics, Danvers, MA). Variant Reporter (Applied Biosystems, Foster City, CA) and Sequencher (Gene Codes, Ann Arbor, MI) softwares were used for analyzing the sequence chromatograms and variant detection.

## Statistical and bioinformatic analysis

Haploview software (www.broadinstitute.org/haploview) was used to calculate allele frequencies. Departure from Hardy-Weinberg equilibrium and difference of allele frequencies between high and low HDL groups were determined using a chi-squared test [35]. A p-value of <0.05 was considered as "suggestive evidence". G power software (version 3.1) was used for power analysis, which showed that our sample size has sufficient power (97% at 5% alpha level) to detect low frequency variants (MAF<0.05) with moderate effect sizes (Cohen's d = 0.05). TagSNPs [(MAF)$\geq$0.05, $r^2$$\geq$0.8)] and linkage disequilibrium (LD) analyses were also performed for common variants [minor allele frequency (MAF) $\geq$5%] in Haploview.

Functional effects of the coding variants were predicted using common *in silico* tools [Sorting Intolerant From Tolerant (SIFT) [36], Mutation Taster 2 (MT2) [37] and Polymorphism Phenotyping v.2 (PP2) [38]. Functional annotations of variants located in non-coding regions were determined using RegulomeDB database (http://regulome.stanford.edu/) and rSNPBase 3.1 database (http://rsnp3.psych.ac.cn) [39, 40]. rSNPBase 3.1 helps to identify SNP-related regulatory elements [transcription factor binding regions (TFBRs), chromatin interactive regions (CIRs), miRNA target sites, lncRNA regions, TADs and circRNAs (circular RNAs)] and their target regulatory genes incorporating mostly experimental data from the Encyclopaedia of DNA Elements (ENCODE) project and other sources such as LNCipedia, CircNet, miRBase, miRNAda and TargetScan. It also provides information related eQTL (expression Quantitative Trait Loci) and disease associations for each query variant. Data annotations for epigenetic marks including active chromatin state regions, histone binding regions, and methylation sites are not available in rSNPBase 3.1, thus we also used RegulomeDB database which uses a scoring scheme to annotate variants based on their regulatory impact related epigenetic marks as well as other regulatory information.

## Results

### DNA sequencing

We identified a total of 412 variants by resequencing the *LIPC* gene in 190 individuals from two distinct population; of which 122 [4 indel, 1 triallelic single nucleotide variant (SNV)

(rs7171818) and 65 diallelic SNVs] were shared in both populations, 56 (4 indels, 152 SNVs) were unique to NHWs and 234 (9 indels, 225 SNVs) were unique to ABs (S2 and S3 Tables). A total of 43 novel variants (not previously reported in any public databases) were observed; of which 26 were present only in ABs and 14 were found only in NHWs. Novel variants were submitted to dbSNP database using handle ID: KAMBOH and distinct dbSNP IDs were assigned for each novel variant (http://www.ncbi.nlm.nih.gov/SNP/snp_viewTable.cgi?handle= KAMBOH). While majority of the identified variants were located in introns, 17 were present in the coding regions of the *LIPC* gene (Fig 1).

Of the 178 variant identified in NHWs, 86 were common (MAF≥0.05), 21 were uncommon (0.01≤MAF<0.05) and 71 were rare (MAF<0.01). Among the 356 variants found in ABs, 172 were common (MAF≥0.05), 116 were uncommon (0.01≤MAF<0.05) and 68 were rare (MAF<0.01) (Fig 2). MAFs of all 178 variants in the total NHW samples and between the two extreme HDL-C groups are shown in S2 Table. Likewise, the same information for the 356 variants in ABs, is shown in S3 Table.

We identified 17 coding SNVs in both populations; of which 11 were synonymous and 6 were non-synonymous. The 6 non-synonymous variants were: valine to methionine (V95M) in exon 3, asparagine to serine (N215S) in exon 5, serine to phenylalanine (S289F) in exon 6, valine to isoleucine (V342I) in exon 6, phenylalanine to leucine (F356L) in exon 7, threonine to methionine (T405M) in exon 8.

## TagSNP identification and linkage disequilibrium (LD) analyses

By using Haploview software, we identified tagSNPs and determined LD between common variants (MAF≥0.05, $r^2$≥0.80) in NHWs and ABs. Of the 174 common variants in ABs (including 3 alleles of the triallelic variant rs7171818, See S3), 3 deviated from Hardy-Weinberg equilibrium (HW-P-value<0.0001) and 10 had low call rate (<80%). Of the 88 common variants in NHWs (including 3 alleles of the triallelic variant rs7171818, See S2), 9 had low call rate (<80%) and 1 deviated from Hardy-Weinberg equilibrium (HW-P-value<0.0003). Accordingly, LD structures for 78 variants in NHWs and 161 variants in ABs were analyzed and tagSNPs were identified by Tagger (see S4 and S5 Tables). Among 78 common variants in NHWs and 161 common variants in ABs, 58 biallelic variants were shared between the two populations. Fig 3 displays the pairwise LD structure of common *LIPC* variants which were highly different in NHWs and ABs.

## Distribution of *LIPC* variants in NHWs and ABs with extreme HDL levels

We analyzed the MAF distributions of quality control (QC) passed 78 common variants in NHWs and 161 variants in ABs between the two extreme HDL-C groups (see S2 and S3

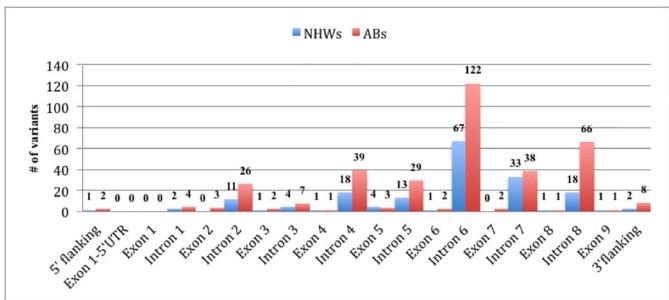

**Fig 1. Locations of the *LIPC* variants identified in NHWs (n = 95) and ABs (n = 95).**

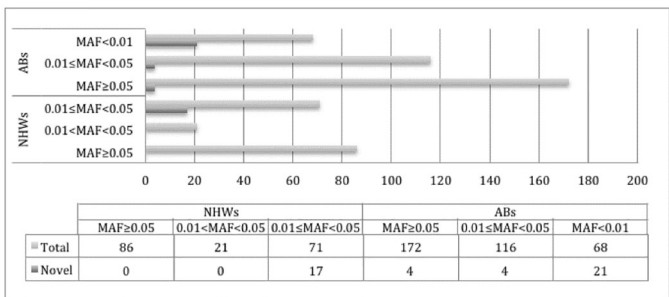

| | | NHWs | | | ABs | |
|---|---|---|---|---|---|---|
| | MAF≥0.05 | 0.01<MAF<0.05 | 0.01≤MAF<0.05 | MAF≥0.05 | 0.01≤MAF<0.05 | MAF<0.01 |
| Total | 86 | 21 | 71 | 172 | 116 | 68 |
| Novel | 0 | 0 | 17 | 4 | 4 | 21 |

**Fig 2. Distributions of the minor allele frequencies of *LIPC* variants in NHWs (n = 95) and ABs (n = 95).**

Tables). Two variants [rs6082 (*P* = 0.0136), rs12592139 (*P* = 0.046)] in ABs (see S3 Table) and one variant [rs143731122 (*P* = 0.028)] in NHWs (see S2 Table) show suggestive evidence for different MAF distributions between extreme HDL-C groups. MAFs of these three variants were lower in the low HDL-C group compared to the high HDL-C group. There were also some variants in both populations that were found either in individuals with the low HDL-C group or in the high HDL-C group. In NHWs, 78 variants (0.005≤MAF≤0.022) were present in one or the other group, including 53 only in low HDL-C group and 25 in only high HDL-C group (S2 Table). In ABs, a total of 89 variants (0.005≤MAF≤0.039) were observed in one or the other group, including 47 in low HDL-C group and 42 in high HDL-C group (S3 Table).

## Functional annotations of *LIPC* variants

Of the 17 *LIPC* coding variants identified, 6 were associated with amino acid changes and 11 were synonymous. SIFT and PP2 tools only predict the effects of non-synonymous variants, however, MT2 evaluates the possible effects of both synonymous and non-synonymous variants. Two rare (MAF = 0.005) *LIPC* variants [rs121912502 (p.Ser289Phe), rs113298164 (p.Thr405Met)], which were unique to NHW population and seen in only individuals with high HDL-C levels were found to be damaging, probably damaging and disease causing in SIFT, PP2 and MT2, respectively (Table 2). The locations of the missense variants on protein structure of the *LIPC* were shown in S2–S7 Figs and MAFs of all identified coding *LIPC* variants reported in gnomAD were listed in Table 2.

Moreover, 4 synonymous *LIPC* variants that were observed in only AB individuals were predicted as disease causing by MT2 computational predictions. Among them, one [rs113174258 (p.Thr71Thr) was a rare variant (MAF = 0.005) and observed in only one individual with low HDL-C level (29.3mg/dl).

Regulatory relevance of the intronic variants using rSNPbase and RegulomeDB databases revealed several regulatory variants (rSNP) that have potential to affect the binding of regulatory elements. A total of 37 and 46 rSNPs were identified in NHWs and ABs, respectively, that have putative binding regions for circRNA or lncRNA. Among the 37 rSNP identified in NHWss, 23 were unique to one extreme HDL-C group and they were found to be located in circRNA and lncRNA binding regions. Of the 46 rSNPs identified in ABs, 9 were seen in only one extreme HDL-C group with MAF≤0.016 and all were determined to be located in circRNA regions and had RegulomeDB score≥3a indicating their regulatory impact (Table 3).

We also assessed the regulatory features of all *LIPC* variants by submitting all refSNP IDs to the RegulomeDB database (v 2.0) and variants with available data were assigned a score ranging from 1 to 6. S2 and S3 Tables list the RegulomeDB score for all variants identified in NHWs and ABs, respectively. Lower score 1 indicates higher evidence that the variant may

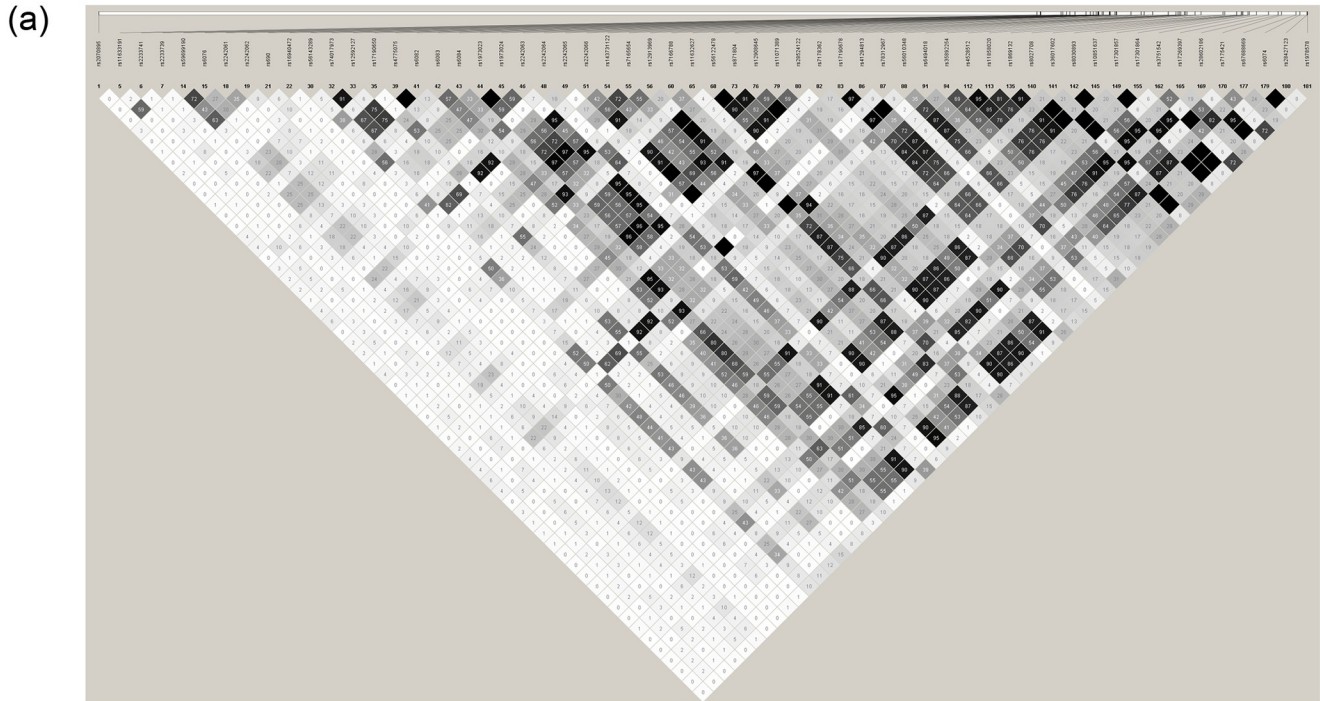

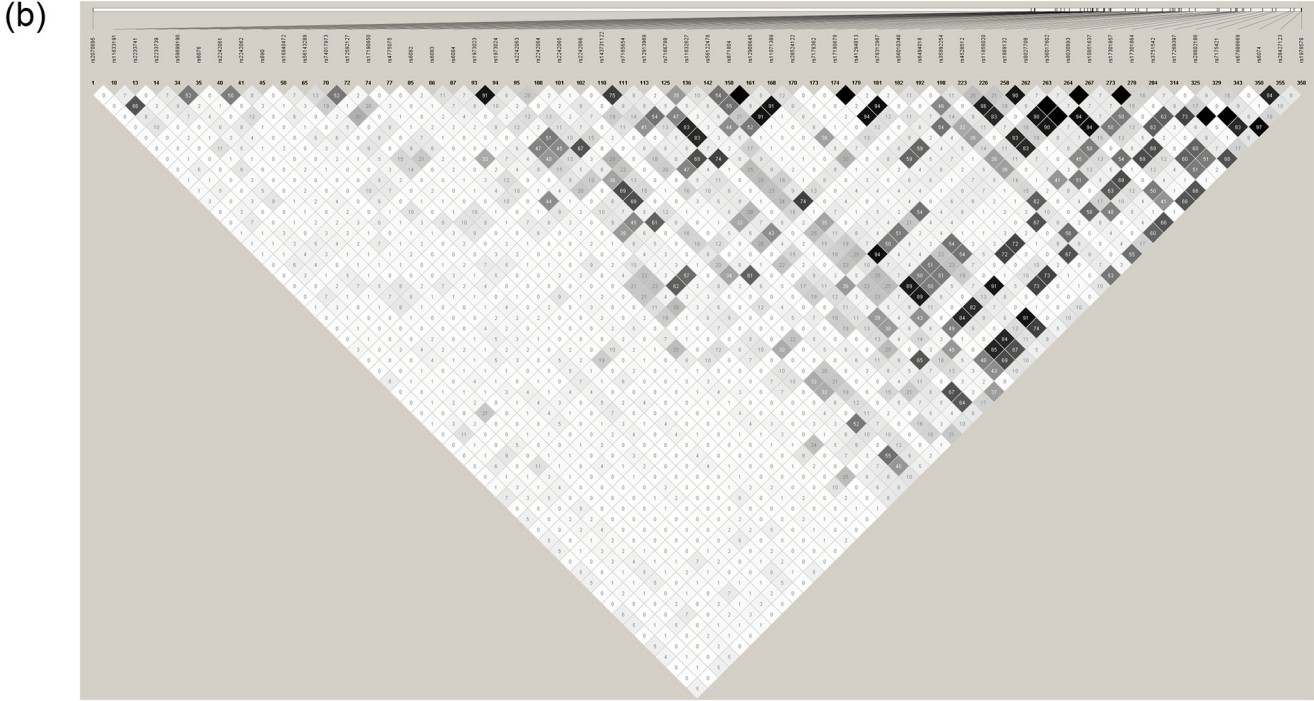

**Fig 3. LD structure of the *LIPC* variants identified in both populations.** (a) NHWs, (b) African blacks. Shade intensity indicates the degree of LD ($r^2$ between 0 and 1). Black indicates complete LD ($r^2 = 1$), white indicates no LD ($r^2 = 0$).

**Table 2. *LIPC* coding variants identified in 95 NHWs and 95 African blacks with extreme HDL-C levels.**

| Ref SNP ID | Location | Amino acid change | African blacks | | | | NHWs | | | | | | | gnomAD (population)* |
|---|---|---|---|---|---|---|---|---|---|---|---|---|---|---|
| | | | Alleles | High HDL-C MAF | Low HDL-C MAF | MAF | Alleles | High HDL-C MAF | Low HDL-C MAF | MAF | SIFT | PP2 | MT | MAF |
| **rs755990193** | Exon 2 | p.T44T | G>A | 0 | 0.011 | 0.005 | G>A | - | - | - | | | P | 1.09E-04 (Asian) |
| rs113174258 | Exon 2 | p.T71T | G>A | 0 | 0.011 | 0.005 | G>A | - | - | - | | | DC | 7.18E-04 |
| rs7175412 | Exon 2 | p.H88H | C>T | 0.042 | 0.087 | 0.064 | C>T | - | - | - | | | DC | 0.004 |
| **rs6078** | **Exon 3** | **p.V95M** | **G>A** | **0.076** | **0.06** | **0.067** | **G>A** | **0.011** | **0** | **0.005** | **T** | **B** | **P** | **0.07** |
| rs776118661 | Exon 3 | p.H127H | C>T | 0.011 | 0.00 | 0.006 | C>T | - | - | - | | | P | **2.01E-05** |
| rs690 | Exon 4 | p.V155V | T>G | 0.5 | 0.44 | 0.473 | T>G | 0.5 | 0.362 | 0.428 | | | P | 0.5 |
| rs6082 | Exon 5 | p.G197G | A>G | 0.115 | 0.02 | 0.07 | A>G | 0.076 | 0.064 | 0.07 | | | P | 0.459 |
| **rs6083** | **Exon 5** | **p.N215S** | **G>A** | **0.219** | **0.244** | **0.231** | **A>G** | **0.38** | **0.34** | **0.36** | **T** | **B** | **P** | **0.474** |
| rs6084 | Exon 5 | p.T224T | C>G | 0.188 | 0.200 | 0.194 | G>C | 0.424 | 0.5 | 0.462 | | | P | 0.424 |
| rs146299102 | Exon 6 | p.H279H | C>T | 0.021 | 0.022 | 0.022 | C>T | - | - | - | | | DC | **0.001** |
| **rs121912502** | **Exon 6** | **p.S289F** | **C>T** | - | - | - | **C>T** | **0.011** | **0** | **0.005** | **D** | **PD** | **DC** | **0.001** |
| **rs145811475** | **Exon 6** | **p.V342I** | **G>A** | **0** | **0.011** | **0.005** | **G>A** | - | - | - | **T** | **B** | **P** | **1.06E-04** |
| **rs3829462** | **Exon 7** | **p.F356L** | **A>C** | **0.032** | **0.03** | **0.031** | **A>C** | - | - | - | **T** | **B** | **P** | **0.031** |
| rs3829461 | Exon 7 | p.T366T | G>A | 0.032 | 0.03 | 0.031 | G>A | - | - | - | | | P | 0.031 |
| **rs113298164** | **Exon 8** | **p.T405M** | **C>T** | - | - | - | **C>T** | **0.011** | **0** | **0.005** | **D** | **PD** | **DC** | **0.003** |
| rs75983069 | Exon 8 | p.P438P | A>G | 0.032 | 0.02 | 0.027 | A>G | - | - | - | T | | DC | **7.46E-04** |
| rs6074 | Exon 9 | p.T479T | C>A | 0.062 | 0.085 | 0.074 | C>A | 0.174 | 0.104 | 0.138 | T | | P | 0.188 |

T: Tolerated, B: Benign, PD: Probably damaging, DC: Disease causing, P: Polymorphism, Bold indicates non-synonymous variants.

*Study-wide MAF is given if variant is found more than one population in the gnomAD (Genome Aggregation Database).

affect the binding region of regulatory elements and have impact on the expression of the target of the gene, whereas score 2 indicates strong evidence that the variant may reside in the binding site of the regulatory elements and proteins. RegulomeDB analyses revealed 36 variants in ABs and 20 variants in NHWs with strong evidence (RegulomeDB score<3) indicating their regulatory roles, and some were also identified as rSNP based on rSNPbase results (S2 and S3 Tables).

## Discussion

Accumulating evidence supports the influence of genetic variants on plasma concentration of HDL-C levels. Majority of the reported genetic variants associated with plasma HDL-C levels reside in genes coding for key enzymes in lipid metabolism. *LIPC* is one of the key lipid genes where its common sequence variants have been reported to be associated with variation in lipoprotein-lipid levels and CAD risk [41–45]. Moreover, individuals with rare loss of functional variants in the *LIPC* gene were observed to have deficient hepatic lipase, resulting in lipid disturbance and affecting CHD risk [46].

In order to further understand the role of common and rare *LIPC* variants in affecting plasma HDL-C levels, we resequenced the *LIPC* gene in individuals with extreme HDL-C levels in two ethnic groups and identified 412 variants, including 290 population-specific and several rare variants with MAF<0.01 (Fig 2), of which some were only present in one of the two extreme HDL-C groups (see S2 and S3 Tables). Our LD analyses showed population-specific LD structure of common *LIPC* variants where there was high LD between variants in NHWs compared to ABs (Fig 3).

**Table 3. Distributions of the regulatory variants (rSNPs) identified in 190 individuals (95 NHWs and 95 ABs) with extreme HDL-C levels.**

| RefSNP ID | Alleles | Location | Call rate | HW-*P* | High HDL-C MAF | Low HDL-C MAF | Total MAF | RegulomeDB Score | Related regulatory elements |
|---|---|---|---|---|---|---|---|---|---|
| | | | | | non-Hispanic Whites (n = 95) | | | | |
| rs143186931 | A>G | Intron 7 | 95.8 | 1 | 0.023 | 0 | 0.011 | 5 | circRNA region |
| rs11071390 | A>G | Intron 7 | 95.8 | 1 | 0 | 0.011 | 0.005 | 5 | circRNA region |
| rs144831345 | G>A | Intron 7 | 95.8 | 1 | 0 | 0.011 | 0.005 | 5 | circRNA region |
| rs117852639 | C>A | Intron 7 | 95.8 | 1 | 0 | 0.021 | 0.011 | 5 | circRNA region |
| rs113298164 | C>T | Exon 8 | 96.8 | 1 | 0.011 | 0 | 0.005 | 5 | circRNA region |
| rs11631342 | A>G | Intron 1 | 97.9 | 1 | 0.022 | 0 | 0.011 | 4 | lncRNA region |
| rs4774305 | G>C | Intron 7 | 97.9 | 1 | 0 | 0.01 | 0.005 | 4 | circRNA region |
| rs139878091 | A>G | Intron 7 | 98.9 | 1 | 0.011 | 0 | 0.005 | 5 | circRNA region |
| rs117911817 | G>A | Intron 7 | 98.9 | 1 | 0.033 | 0 | 0.016 | 5 | circRNA region |
| rs1869129 | T>C | Intron 7 | 100 | 1 | 0 | 0.01 | 0.005 | 3a | circRNA region |
| rs1869130 | C>T | Intron 7 | 100 | 1 | 0 | 0.01 | 0.005 | 4 | circRNA region |
| rs12438032 | G>A | Intron 7 | 100 | 1 | 0 | 0.01 | 0.005 | 5 | circRNA region |
| rs34964641 | T>G | Intron 7 | 100 | 1 | 0 | 0.01 | 0.005 | 5 | circRNA region |
| rs35925692 | Ins1 | Intron 7 | 100 | 1 | 0 | 0.01 | 0.005 | 5 | circRNA region |
| rs1869131 | A>T | Intron 7 | 100 | 1 | 0 | 0.01 | 0.005 | 5 | circRNA region |
| rs4775079 | C>T | Intron 7 | 100 | 1 | 0 | 0.01 | 0.005 | 5 | circRNA region |
| rs8026372 | A>G | Intron 7 | 100 | 1 | 0 | 0.01 | 0.005 | 5 | circRNA region |
| rs1839928 | A>G | Intron 7 | 100 | 1 | 0 | 0.01 | 0.005 | 5 | circRNA region |
| rs1839927 | A>G | Intron 7 | 100 | 1 | 0 | 0.01 | 0.005 | 5 | circRNA region |
| rs8030903 | T>C | Intron 7 | 100 | 1 | 0 | 0.01 | 0.005 | 4 | circRNA region |
| rs10851636 | C>T | Intron 7 | 100 | 1 | 0 | 0.01 | 0.005 | 4 | circRNA region |
| rs7170227 | G>A | Intron 7 | 100 | 1 | 0 | 0.01 | 0.005 | 3a | circRNA region |
| rs35412158 | G>A | Intron 7 | 100 | 1 | 0 | 0.01 | 0.005 | 5 | circRNA region |
| | | | | | African Blacks (n = 95) | | | | |
| rs35631005 | C:T | Intron 6 | 97.9 | 1 | 0.011 | 0 | 0.005 | 4 | |
| rs533300601 | G:A | Intron 7 | 97.9 | 0.0324 | 0.033 | 0 | 0.016 | 3a | circRNA region |
| rs16940468 | T:G | Intron 2 | 96.8 | 1 | 0.00 | 0.011 | 0.005 | 5 | circRNA region |
| rs12909325 | G:A | Intron 2 | 96.8 | 1 | 0.00 | 0.021 | 0.011 | 5 | circRNA region |
| rs115408618 | G:C | Intron 7 | 96.8 | 1 | 0.023 | 0 | 0.011 | 3a | circRNA region |
| rs190375050 | C:T | Intron 7 | 98.9 | 1 | 0.00 | 0.021 | 0.011 | 5 | circRNA region |
| rs181084356 | C:G | Intron 7 | 98.9 | 1 | 0.00 | 0.021 | 0.011 | 5 | circRNA region |
| rs568646677 | G:T | Intron 7 | 98.9 | 1 | 0.011 | 0 | 0.005 | 4 | circRNA region |
| rs143889538 | G:A | Intron 7 | 98.9 | 1 | 0.021 | 0 | 0.011 | 2b | circRNA region |

To the best of our knowledge, this is the first study that provides a catalog of common and rare *LIPC* variants by resequencing the entire gene in two well-characterized population based samples with extreme HDL-C levels. In a recent study, only exons and exon-intron junctions of *LIPC* were resequenced in Koreans with extreme HDL-C levels (n = 42) where no rare variants were identified [27]. Three non-synonymous variants (rs3829462, rs6078, rs6083) were identified in Korean subjects with extreme HDL-C levels of which rs3829462 and rs6078 were seen in all studied subjects (n = 42) and rs6083 were found in only 16 individuals. In our study, the rs3829462 SNP was observed only in the AB sample and its distribution along with rs6078 and rs6083 were similar between the two HDL-C groups in ABs. In NHWs, the distribution of rs6083 was also similar between the two extreme groups, and rs6078 was observed in only one individual with high HDL-C (see S3 Table).

Moreover, we found three common variants [rs6082 ($P$ = 0.0136), rs12592139 ($P$ = 0.0463), rs143731122 ($P$ = 0.028)] that showed evidence of association with extreme HDL-C levels (see S3 and S4 Tables). The rs6082 (p.G197G) is a synonymous coding variant that was previously identified in individuals with HL deficiency and is listed as a benign variant in the ClinVar database. The most extensively studied common *LIPC* variants are located in the promoter, -250G/A (rs2070895) and -514C/T (rs1800588), that have been shown to be associated with HL activity, HDL-C and with the risk of metabolic diseases [19, 41, 45, 47–49]. The location of -514C/T (rs1800588) was out of our sequenced region range and thus we did not have data for this variant On the other hand, -250G/A (rs2070895) was observed in both populations and the frequency of the A allele was higher in individuals with high HDL-C group than in low HDL-C group in both NHWs (20.7% vs 12.8%) and ABs (65.6% vs 54.3%). Our results are in good agreement with a recent GWAS finding where the rs2070895-A was associated with increased levels of HDL-C levels ($p$ = 4 x 10$^{-24}$) [50].

We also observed 78 uncommon or rare variants (MAF<0.05) in NHWs and 89 in ABs that were present in only one or the other extreme HDL-C group. Of the variants that were identified in only one extreme group, seven were coding including four [rs6078 (p.V95M), rs121912502 (p.S289F), rs145811475 (p.V342I), rs113298164 (p.T405M) causing amino acid change (Table 2). The rs121912502 (p.Ser289Phe) variant was observed in only one NHW individual in the high HDL-C group and this was reported to be of uncertain significance associated with HL deficiency in ClinVar database. Another coding variant, rs113298164 (p. Thr405Met), which was reported as pathogenic and associated with HL deficiency in ClinVar, was also found in only one NHW individual with high HDL-C levels. Our finding is in accordance with a recent study which suggested the association of the minor T allele of rs113298164 with high HDL-C levels [51]. Our results indicate that the sequencing subset of samples having extreme HDL-C levels selected from the larger population-based samples have sufficient power to detect variants with strong effect size. Notably, our sequencing study design on extreme lipid phenotypes using the same sample size has enabled us to detect multiple rare and novel variants in addition to common variants in other candidate lipid genes [52–58].

Although the effects of coding *LIPC* variants on lipid traits have been widely investigated, the contribution of *LIPC* non-coding regulatory variants in interindividual variation in plasma lipids is still unclear. Thus, we also assessed the regulatory nature of non-coding *LIPC* variants by using two databases (RegulomeDB and rSNPbase) to prioritize the non-coding variants as regulatory variants. None of the identified 43 novel non-coding variants were found to be located in regions that have regulatory significance. However, 42 *LIPC* variants, seen in one or the other extreme HDL-C group, were implicated to reside in binding regions of regulatory elements, including one exonic variant [rs113298164 (p.Thr405Met)]. Our analyses suggest that non-coding regulatory *LIPC* variants have the potential to disrupt the regulatory functions of circRNA and lncRNA as well as highlight the possible regulatory role of exonic rs113298164 variant in splicing dysregulation. The impact of exonic variants in splicing mechanisms and their contribution to phenotypes highlight the imperative role of investigating the roles of exonic variants in splicing dysregulation [59]. Interestingly, majority of the regulatory *LIPC* variants were seen in individuals with the low HDL-C in both populations (Table 3).

Our study has some limitations. Our sequencing did not cover the large intron 1 due to technical reasons and thus we missed the potential rare and common *LIPC* sequence variants in this region. Our results depend on the data of 190 chromosomes with extreme phenotypes in each ethnic group and our sample size is small to claim associations between variants and HDL-C levels. Thus variants with suggestive evidence of association with HDL-C levels should be tested in a larger sample. Nevertheless, our effort produced a large catalog of *LIPC* sequence variants in samples from two ethnic groups and yielded several novel variants as well as rare population-specific

variants located in the *LIPC* regulatory regions that could be followed up in larger future studies. Also, our findings suggest that individuals with extreme HDL-C levels carrying variants that might affect the activity of LIPC should be followed up for their risk cardiovascular disease.

## Conclusions

In conclusion, our study reaffirms the considerable contribution of *LIPC* variants to plasma concentrations of HDL-C levels in the general population and also highlights the high prevalence of the rare variants that may play key role in the regulation of the HDL-C levels. Our results also emphasize the possible regulatory impact of non-coding *LIPC* variants in the determination of HDL-C levels. However, associations between suggested regulatory *LIPC* variants and lipid traits should be tested in large samples and functional studies need to be conducted to further evaluate the roles of non-coding regulatory variants in the lipid metabolism.

## Supporting information

**S1 Fig. Exon and intron structure of the *LIPC* gene.** The image was retrieved from http://www.ensembl.org/. Boxes and lines between boxes indicate exons and introns, respectively. Unfilled box indicates untranslated region.
(TIF)

**S2 Fig. Diagram depicts a part of the hepatic lipase protein structure where rs6078 (p. V95M) is located.** The image was retrieved from the VarSite database (https://www.ebi.ac.uk/thornton-srv/databases/VarSite).
(TIF)

**S3 Fig. Diagram depicts a part of the hepatic lipase protein structure where rs6083 (p. N215S) is located.** The image was retrieved from the VarSite database (https://www.ebi.ac.uk/thornton-srv/databases/VarSite).
(TIF)

**S4 Fig. Diagram depicts a part of the hepatic lipase protein structure where rs121912502 (p.S289F) is located.** The image was retrieved from the VarSite database (https://www.ebi.ac.uk/thornton-srv/databases/VarSite).
(TIF)

**S5 Fig. Diagram depicts a part of the hepatic lipase protein structure where rs145811475 (p. V342I) is located.** The image was retrieved from the VarSite database (https://www.ebi.ac.uk/thornton-srv/databases/VarSite).
(TIF)

**S6 Fig. Diagram depicts a part of the hepatic lipase protein structure where rs145811475 (p.F356L) is located.** The image was retrieved from the VarSite database (https://www.ebi.ac.uk/thornton-srv/databases/VarSite).
(TIF)

**S7 Fig. Diagram depicts a part of the hepatic lipase protein structure where rs145811475 (p. T405M) is located.** The image was retrieved from the VarSite database (https://www.ebi.ac.uk/thornton-srv/databases/VarSite).
(TIF)

**S1 Table. Primers used in DNA sequencing and PCR.**
(DOCX)

**S2 Table. Sequencing results for the *LIPC* gene in NHWs (n = 95).**
(DOCX)

**S3 Table. Sequencing results for the *LIPC* gene in African blacks (n = 95).**
(DOCX)

**S4 Table. Tagger results for 161 *LIPC* variants (MAF≥0.05, r2≥0.8) in ABs.**
(DOCX)

**S5 Table. Tagger results for the 78 *LIPC* variants (MAF≥0.05, r2≥0.8) in NHWs.**
(DOCX)

## Author Contributions

**Conceptualization:** Dilek Pirim, M. Ilyas Kamboh.

**Data curation:** Dilek Pirim.

**Formal analysis:** Dilek Pirim, M. Ilyas Kamboh.

**Funding acquisition:** M. Ilyas Kamboh.

**Investigation:** Dilek Pirim, F. Yesim Demirci, M. Ilyas Kamboh.

**Methodology:** Dilek Pirim, F. Yesim Demirci, M. Ilyas Kamboh.

**Project administration:** F. Yesim Demirci, M. Ilyas Kamboh.

**Resources:** Clareann H. Bunker, John E. Hokanson, Richard F. Hamman, M. Ilyas Kamboh.

**Supervision:** F. Yesim Demirci, M. Ilyas Kamboh.

**Validation:** Dilek Pirim, M. Ilyas Kamboh.

**Visualization:** M. Ilyas Kamboh.

**Writing – original draft:** Dilek Pirim, M. Ilyas Kamboh.

**Writing – review & editing:** Dilek Pirim, Clareann H. Bunker, John E. Hokanson, Richard F. Hamman, F. Yesim Demirci, M. Ilyas Kamboh.

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
