## [Decision Letter · Decision Letter 0]

19 Oct 2020

PONE-D-20-29487

Hepatic Lipase (LIPC) sequencing in individuals with extremely high and low high-density lipoprotein cholesterol levels

PLOS ONE

Dear Dr. Pirim,

Thank you for submitting your manuscript to PLOS ONE. After careful consideration, we feel that it has merit but does not fully meet PLOS ONE’s publication criteria as it currently stands. Therefore, we invite you to submit a revised version of the manuscript that addresses the points raised during the review process.

ACADEMIC EDITOR: The paper is interesting, with good methodologies. The reviewers have raised some points of concern, that the authors must address in their rebuttal.

We look forward to receiving your revised manuscript.

Kind regards,

Marco Giorgio Baroni, MD, PhD

Academic Editor

PLOS ONE

Journal Requirements:

"This study was supported by the National Heart, Lung and Blood Institute (NHLBI) grant, HL084613 (M. Ilyas Kamboh). ".

i) Please provide an amended statement that declares *all* the funding or sources of support (whether external or internal to your organization) received during this study, as detailed online in our guide for authors at http://journals.plos.org/plosone/s/submit-now.  Please also include the statement “There was no additional external funding received for this study.” in your updated Funding Statement.

ii) Please include your amended Funding Statement within your cover letter. We will change the online submission form on your behalf.

Reviewers' comments:

Reviewer's Responses to Questions

**Comments to the Author**

1. Is the manuscript technically sound, and do the data support the conclusions?

Reviewer #1: Yes

Reviewer #2: Yes

2. Has the statistical analysis been performed appropriately and rigorously? 

Reviewer #1: Yes

Reviewer #2: Yes

3. Have the authors made all data underlying the findings in their manuscript fully available?

Reviewer #1: Yes

Reviewer #2: Yes

4. Is the manuscript presented in an intelligible fashion and written in standard English?

Reviewer #1: Yes

Reviewer #2: Yes

5. Review Comments to the Author

Reviewer #1: In this study, Pirim D and colleagues performed a resequencing of the LIPC gene in 95 non-Hispanic Whites (NHWs) and 95 African blacks (ABs) selected from the upper and lower 10th percentile of HDL cholesterol distribution. The aim of the study was to identify and understand the role of common, uncommon and rare variants that may influence plasma HDL-C levels. They identified a total of 464 variants, including 43 novel. To assess their functional relevance, in silico functional analyses were executed. Two nonsynonymous variants (p.S289F, p.T405M), found in NHWs with higher HDL-C levels were predicted to have damaging effect on LIPC protein. Furthermore, the authors found several non-coding variants that possibly reside in the circRNA and lncRNA binding sites and may have regulatory function. The authors conclude that this study highlights the importance of LIPC polymorphisms, common and rare, in influencing plasma HDL-C levels and that functional studies are needed to further evaluate the roles of non-coding regulatory variants in the lipid metabolism.

The results presented in the paper are well written and easy to follow and provide additional evidences on the importance of the LIPC gene polymorphisms in the modulation of plasma HDL-C levels.

Nonetheless, I would have some comments to the authors.

1) It’s not clear to me why the authors state that they identified a total of 464 variants if summing the 122 shared in both populations, the 156 unique to NHWs and the 234 unique to Abs, the variants are 512 in total.

2) I found a discrepancy between S2 Table where are reported 180 variants in NHWs of which 88 are common and the test in page 8 “Of the 178 variant identified in NHWs, 86 were common (MAF≥0.05)….”. In paragraph 3.2 (line 4) the authors state they found 88 common variants.

And the same for S3 Table: 358 variants of which 174 common in AB cohort and in the test on page 8 “Among the 356 variants found in ABs, 172 were common (MAF≥0.05)……”

Could the authors explain the differences?

3) in paragraph 3.2, it’s not clear why 161 variants in AB cohort were analysed (S4 Table) if the common variants were 172. I wonder if the 3 variants deviating from H-W equilibrium and the 10 with low call rate have been eliminated. However, in this case the variants analysed in S4 Table should be 159. This should be clarified.

4) I’ve a doubt how the authors calculated the RegulomeDB Score in Table 3. Indeed, I tried to calculate the Score selecting randomly few snps. However, the results are different from those reported in table 3. May the authors explain these different results? I wonder if the RegulomeDB scores calculated in S2 and S3 Tables present the same discrepancy.

NHWs

RefSNP ID RegulomeDB rank

rs117911817 5

rs1869129 3a

rs1869130 4

rs12438032 5

rs34964641 5

rs35925692 5

rs1869131 5

rs4775079 5

ABs

RefSNP ID RegulomeDB rank

rs35631005 4

rs533300601 3a

rs16940468 5

rs12909325 5

rs115408618 3a

rs190375050 5

rs181084356 5

rs568646677 4

rs143889538 2b

Reviewer #2: Pirim and collaborator sequenced LIPC gene in 190 subjects selected from the upper and lower 10th percentile of HDL levels in two cohorts, African Blacks and non-Hispanic Whites. 464 variants were recorded, 43 of which were novel. SNPs were evaluated using some predictive online software for coding and non-coding (regulatory) variants. Among all, 17 coding variants were identified: 6 cause aminoacid change while 11 were synonymous. Authors analysis were focused on difference in MAF distribution within the extremes of HDL levels. Overall, 3 variants showed a significant different distribution. In particularly, MAF were lower in low-HDL than high-HDL group. Moreover, authors found 3 common variants associated with extremes of HDL levels. Other findings are in line with previous published studies.

Language and exposition are very clear. Also the amount of data, figures, tables and supplementary materials provided make easy to follow authors argument. Interestingly, the analysis considers cohorts from different ethnic group, non-Hispanic White and African Black.

Minor revision:

Lacks of power calculation (even if cited at page 15). As it would be useful for readers and for fully understand study design, could authors add a power calculation in statistical analysis paragraph, 2.4?

Conclusion paragraphs seems too short. Could authors add a brief summary of findings, or most promising variants, or some hypothesis for observed effects in coding and regulatory variants, or some future prospective of the study?

at page 8. On paragraph 3.1 the common variants found in NWHs are n=86, while in the same page, but in paragraph 3.2, are n=88. Please correct or explain.

6. PLOS authors have the option to publish the peer review history of their article (what does this mean?). If published, this will include your full peer review and any attached files.

Reviewer #1: No

Reviewer #2: No

---

## [Author Response · Author response to Decision Letter 0]

19 Nov 2020

Dear Editors:

Thank you for the opportunity to revise our manuscript. We are pleased with the overall positive review and found the comments very helpful in improving our manuscript. We have revised the manuscript accordingly (highlighted in red in the manuscript) and below we provide a point-by-point response to each comment in red. 

The Funding Statement has been also updated as "This study was supported by the National Heart, Lung and Blood Institute (NHLBI) grant, HL084613 (M. Ilyas Kamboh). The funders had no role in study design, data collection and analysis, decision to publish, or preparation of the manuscript. There was no additional external funding received for this study ".

Reviewer #1:

In this study, Pirim D and colleagues performed a resequencing of the LIPC gene in 95 non-Hispanic Whites (NHWs) and 95 African blacks (ABs) selected from the upper and lower 10th percentile of HDL cholesterol distribution. The aim of the study was to identify and understand the role of common, uncommon and rare variants that may influence plasma HDL-C levels. They identified a total of 464 variants, including 43 novel. To assess their functional relevance, in silico functional analyses were executed. Two nonsynonymous variants (p.S289F, p.T405M), found in NHWs with higher HDL-C levels were predicted to have damaging effect on LIPC protein. Furthermore, the authors found several non-coding variants that possibly reside in the circRNA and lncRNA binding sites and may have regulatory function. The authors conclude that this study highlights the importance of LIPC polymorphisms, common and rare, in influencing plasma HDL-C levels and that functional studies are needed to further evaluate the roles of non-coding regulatory variants in the lipid metabolism.

The results presented in the paper are well written and easy to follow and provide additional evidences on the importance of the LIPC gene polymorphisms in the modulation of plasma HDL-C levels.

Nonetheless, I would have some comments to the authors.

Response: We greatly appreciate the reviewer for noting that our manuscript is well written, easy to follow and the importance of novel findings. 

1) It’s not clear to me why the authors state that they identified a total of 464 variants if summing the 122 shared in both populations, the 156 unique to NHWs and the 234 unique to Abs, the variants are 512 in total.

Response: We apologize for this typo as the number of variants unique to NHWs should have been 56 not "156", so the total number of identified variants is 412. Abstract, results and discussion sections have now been edited accordingly.

2) I found a discrepancy between S2 Table where are reported 180 variants in NHWs of which 88 are common and the test in page 8 “Of the 178 variant identified in NHWs, 86 were common (MAF≥0.05)….”. In paragraph 3.2 (line 4) the authors state they found 88 common variants.

And the same for S3 Table: 358 variants of which 174 common in AB cohort and in the test on page 8 “Among the 356 variants found in ABs, 172 were common (MAF≥0.05)……”

Could the authors explain the differences?

Response: The discrepancy was due to a triallelic variant (rs7171818) that was found in both populations. Including the triallelic variant, the total number of common variants identified in NHWs and ABs were 86 and 172, respectively. However, since the three alleles for the rs7171818 were presented in 3 separate rows, the number of total variants falsely got added to 88 and 174 in NHWs and ABs, respectively. In order to avoid this apparent mis-representation, now the triallelic variant in Table S2 and S3 is presented in a single row. 

Of note, Haploview analyses was conducted for all variants, including three alleles for the triallelic variant. For further clarification, we have edited the text on Page 8, Section "TagSNP identification and Linkage Disequilibrium (LD) analyses", line 2-6 as follows:

“Of the 174 common variants in ABs (including 3 alleles of the triallelic variant rs7171818, See S3), 3 deviated from Hardy-Weinberg equilibrium (HW-P-value<0.0001) and 10 had low call rate (<80%). Of the 88 common variants in NHWs (including 3 alleles of the triallelic variant rs7171818, See S2), 9 had low call rate (<80%) and 1 deviated from Hardy-Weinberg equilibrium (HW-P-value<0.0003).”

3) in paragraph 3.2, it’s not clear why 161 variants in AB cohort were analysed (S4 Table) if the common variants were 172. I wonder if the 3 variants deviating from H-W equilibrium and the 10 with low call rate have been eliminated. However, in this case the variants analysed in S4 Table should be 159. This should be clarified.

Response: In ABs, 174 common variants (including the triallelic variant rs7171818) were analyzed. "172" was a typo, which is now corrected on Page 8, Section 3.2, line 2-4, as indicated above.

4) I’ve a doubt how the authors calculated the RegulomeDB Score in Table 3. Indeed, I tried to calculate the Score selecting randomly few snps. However, the results are different from those reported in table 3. May the authors explain these different results? I wonder if the RegulomeDB scores calculated in S2 and S3 Tables present the same discrepancy.

NHWs

RefSNP ID RegulomeDB rank

rs117911817 5

rs1869129 3a

rs1869130 4

rs12438032 5

rs34964641 5

rs35925692 5

rs1869131 5

rs4775079 5

ABs

RefSNP ID RegulomeDB rank

rs35631005 4

rs533300601 3a

rs16940468 5

rs12909325 5

rs115408618 3a

rs190375050 5

rs181084356 5

rs568646677 4

rs143889538 2b

Response: Thank you for pointing this out. We used the RegulomeDB version 1 in our analyses and since then the database has been updated, resulting in difference scores for some variants than the previous version. 

We have now reanalyzed all variants by using RegulomeDB v2 and updated all scores in the tables and related text accordingly.

Reviewer #2:

Pirim and collaborator sequenced LIPC gene in 190 subjects selected from the upper and lower 10th percentile of HDL levels in two cohorts, African Blacks and non-Hispanic Whites. 464 variants were recorded, 43 of which were novel. SNPs were evaluated using some predictive online software for coding and non-coding (regulatory) variants. Among all, 17 coding variants were identified: 6 cause aminoacid change while 11 were synonymous. Authors analysis were focused on difference in MAF distribution within the extremes of HDL levels. Overall, 3 variants showed a significant different distribution. In particularly, MAF were lower in low-HDL than high-HDL group. Moreover, authors found 3 common variants associated with extremes of HDL levels. Other findings are in line with previous published studies.

Language and exposition are very clear. Also the amount of data, figures, tables and supplementary materials provided make easy to follow authors argument. Interestingly, the analysis considers cohorts from different ethnic group, non-Hispanic White and African Black.

Response: We greatly appreciate the reviewer for noting multiple strengths in our study. 

Minor revision:

Lacks of calculation (even if cited at page 15). As it would be useful for readers and for fully understand study design, could authors add a power calculation in statistical analysis paragraph, 2.4?

Response: We have now calculated the power by using G power based on our sample size and the text has been revised as below on Page 6, Section Statistical and Bioinformatic Analysis, line 4-6: 

“G power software (version 3.1) was used for power analysis, which showed that our sample size has sufficient power (97% at 5% alpha level) to detect low frequency variants (MAF<0.05) with moderate effect sizes (Cohen's d=0.05).” 

Conclusion paragraphs seems too short. Could authors add a brief summary of findings, or most promising variants, or some hypothesis for observed effects in coding and regulatory variants, or some future prospective of the study?

Response: Since the summary of the findings is already highlighted in the Abstract and the promising variants discussed in the Discussion section, we avoided their repetitions in the Conclusions. However, in deference to the reviewer's suggestion, we have added the the below text in the Conclusions section on Page 16, Section 5, line 3-6:

“Our results also emphasize the possible regulatory impact of non-coding LIPC variants in the determination of HDL-C levels. However, associations between suggested regulatory LIPC variants and lipid traits should be tested in large samples....”

at page 8. On paragraph 3.1 the common variants found in NWHs are n=86, while in the same page, but in paragraph 3.2, are n=88. Please correct or explain.

Response: As responded to under Reviewer 1, in NHWs, we identified 86 common variants, including one triallelic SNP (rs7171818) and for clarification, we have edited the text on Page 8, Section TagSNP identification and Linkage Disequilibrium (LD) analyses, line 2-6.

We hope that we have satisfactorily addressed reviewers’ concerns and look forward to learning a favorable response.

Sincerely,

M. Ilyas Kamboh, PhD, FAHA

---

## [Decision Letter · Decision Letter 1]

1 Dec 2020

Hepatic Lipase (LIPC) sequencing in individuals with extremely high and low high-density lipoprotein cholesterol levels

PONE-D-20-29487R1

Dear Dr. Pirim,

We’re pleased to inform you that your manuscript has been judged scientifically suitable for publication and will be formally accepted for publication once it meets all outstanding technical requirements.

Kind regards,

Marco Giorgio Baroni, MD, PhD

Academic Editor

PLOS ONE

Additional Editor Comments (optional):

Reviewers' comments:

Reviewer's Responses to Questions

**Comments to the Author**

1. If the authors have adequately addressed your comments raised in a previous round of review and you feel that this manuscript is now acceptable for publication, you may indicate that here to bypass the “Comments to the Author” section, enter your conflict of interest statement in the “Confidential to Editor” section, and submit your "Accept" recommendation.

Reviewer #1: All comments have been addressed

Reviewer #2: All comments have been addressed

2. Is the manuscript technically sound, and do the data support the conclusions?

Reviewer #1: Yes

Reviewer #2: Yes

3. Has the statistical analysis been performed appropriately and rigorously? 

Reviewer #1: Yes

Reviewer #2: Yes

4. Have the authors made all data underlying the findings in their manuscript fully available?

Reviewer #1: Yes

Reviewer #2: Yes

5. Is the manuscript presented in an intelligible fashion and written in standard English?

Reviewer #1: Yes

Reviewer #2: Yes

6. Review Comments to the Author

Reviewer #1: Dear authors,

I read the paper carefully and the current form is suitable to be published. I really appreciated your collaboration in improving your manuscript.

Reviewer #2: (No Response)

7. PLOS authors have the option to publish the peer review history of their article (what does this mean?). If published, this will include your full peer review and any attached files.

Reviewer #1: No

Reviewer #2: No

---

## [Editor Report · Acceptance letter]

4 Dec 2020

PONE-D-20-29487R1 

Hepatic Lipase (*LIPC*) sequencing in individuals with extremely high and low high-density lipoprotein cholesterol levels 

Dear Dr. Pirim:

I'm pleased to inform you that your manuscript has been deemed suitable for publication in PLOS ONE. Congratulations! Your manuscript is now with our production department. 

Kind regards, 

on behalf of

Prof Marco Giorgio Baroni 

Academic Editor

PLOS ONE